# Anticipated and experienced stigma and discrimination in the workplace among individuals with major depressive disorder in 35 countries: qualitative framework analysis of a mixed-method cross-sectional study

Tine Van Bortel [1,2] Nuwan Darshana Wickramasinghe [3] Samantha Treacy,[4] Nashi Khan,[5] Uta Ouali,[6] Athula Sumathipala,[7,8] Vesna Svab,[9] Doaa Nader,[10] Nadia Kadri,[11] Maria Fatima Monteiro,[12] Lee Knifton,[13] Neil Quinn,[14] Chantal Van Audenhove,[15] Antonio Lasalvia,[16] Chiara Bonetto,[16] Graham Thornicroft,[17] Jaap van Weeghel,[18,19] Evelien Brouwers,[18,20] ASPEN-INDIGO Study Group Credit

For numbered affiliations see end of article.

**Correspondence to**
Professor Tine Van Bortel;
tv250@cam.ac.uk

## ABSTRACT

**Objectives** Workplace stigmatisation and discrimination are significant barriers to accessing employment opportunities, reintegration and promotion in the workforce for people with mental illnesses in comparison to other disabilities. This paper presents qualitative evidence of anticipated and experienced workplace stigma and discrimination among individuals with major depressive disorder (MDD) in 35 countries, and how these experiences differ across countries based on their Human Development Index (HDI) level.

**Design** Mixed-method cross-sectional survey.

**Participants, setting and measures** The qualitative data were gathered as part of the combined European Union Anti-Stigma Programme European Network and global International Study of Discrimination and Stigma Outcomes for Depression studies examining stigma and discrimination among individuals with MDD across 35 countries. Anticipated and experienced stigma and discrimination were assessed using the Discrimination and Stigma Scale version 12 (DISC-12). This study used responses to the open-ended DISC-12 questions related to employment. Data were analysed using the framework analysis method.

**Results** The framework analysis of qualitative data of 141 participants identified 6 key 'frames' exploring (1) participants reported experiences of workplace stigma and discrimination; (2) impact of experienced workplace stigma and discrimination; (3) anticipated workplace stigma and discrimination; (4) ways of coping; (5) positive work experiences and (6) contextualisation of workplace stigma and discrimination. In general, participants from very high HDI countries reported higher levels of anticipated and experienced discrimination than other HDI groups (eg, less understanding and support, being more avoided/shunned, stopping themselves from looking for work because of expectation and fear of discrimination). Furthermore, participants from medium/low HDI countries were more likely to report positive workplace experiences.

**Conclusions** This study makes a significant contribution towards workplace stigma and discrimination among individuals with MDD, still an under-researched mental health diagnosis. These findings illuminate important relationships that may exist between countries/contexts and stigma and discrimination, identifying that individuals from very high HDI countries were more likely to report anticipated and experienced workplace discrimination.

## STRENGTHS AND LIMITATIONS OF THIS STUDY

⇒ Large global study across 35 countries investigating workplace stigma and discrimination of individuals with major depressive disorder, which is still a very under-researched condition.

⇒ The overall mixed-method global study, and this qualitative framework substudy, are currently the first ones to examine whether and how the Human Development Index (HDI) level is associated with experienced or anticipated stigma and discrimination, which is innovative and significant.

⇒ Approximately one-fourth of available interviews were not included due to limited information regarding workplace issues.

⇒ While having a large study sample from 35 countries provides useful insights across different HDI countries, the proportion from each country is smaller; therefore, these findings should be viewed as exploratory with more in-depth local research needed.

## INTRODUCTION

In spite of the positive strides taken towards fostering more egalitarian working environments during the past few decades, global evidence suggests that workplace stigma and discrimination against people with mental illness continue to be an important issue that has not been addressed adequately.[1 2] Workplace stigmatisation and discrimination have been identified as significant hindering factors in accessing employment opportunities, reintegration and promotion in the workforce for people with mental illnesses in comparison to other disabilities.[3–18] The literature highlights that people with mental illnesses experience direct discrimination due to prejudicial attitudes from employers as well as workmates[4 5] while historical patterns of disadvantage, structural disincentives against competitive employment and generalised policy neglect lead to indirect discrimination.[12 13]

Sociocultural factors also play a role in mental ill health stigma and discrimination, including in the workplace.[19–23] However, the general literature in this respect remains scarce, especially pertaining to the workplace. In their cultural comparison of mental illness stigma and help-seeking attitudes, Fekih-Romdhane *et al*[20] found that very limited literature is available and mostly focused on high-income countries (HICs) rather than on low-income and middle-income countries (LMICs). These studies also indicate that mental illness stigma is more disorder-specific across HICs. For instance, several studies showed that people in Japan hold more unfavourable opinions about persons with schizophrenia than those from other countries, including the UK, Australia, Taiwan and Indonesia. Furthermore, they more strongly desire to distance themselves from a person with mental illness. Consequently, institutional care for mentally ill patients is often favoured. They also found that, in HICs, the awareness about mental illness and related stigma is better than in LMICs; yet, stigmatising attitudes persist in the community and in the workplace. In HICs, there is also a more stigmatising attitude towards more severe mental illness and disability. This seems to be less prominent in non-industrialised societies due to a more supportive environment with more social cohesion and therefore less risk of prolonged rejection, isolation, segregation and institutionalisation.[20 21]

Similarly, in their comparison study between Arab and Western countries, Vaishnav *et al*[19] found cultural and cross-national differences too. They concluded that substantial stigma towards mental illness also exists in various Arab countries, but that Arab people are more likely to approve of psychosocial causation than Western people; with significant differences in cultural beliefs of mental health problems according to the Arab country of origin. An inclination towards culturally related beliefs regarding mental health causation—such as God's punishment, evil spirits, demons and black magic—has been documented in Arab as compared with non-Arab populations, with substantial cross-cultural variations between the different Arab communities as well.[19]

Sociocultural factors also play a critical role in the onset of stigma, its perpetuation and the success of antistigma measures. Literature suggests that forms and extent of stigma in Asian countries are different from the Western world and that these are not homogeneous entities either. However, there are limited data from Asia and LMICs concerning stigma, including antistigma measures.[20]

It is clear that mental ill health stigma and discrimination are prevalent worldwide; however, its determinants and manifestations can vary from country to country and across cultural contexts.[19–23] In their cross-national variations of stigma and discrimination study, Lasalvia *et al* found that people living in very high Human Development Index (HDI) countries reported higher discrimination than those in medium/low HDI countries. Cross-countries variations in discrimination were only partially explained by individual-level variables. Contextual factors clearly play an important role. Therefore, country-specific and context-specific interventions should be implemented and more research should be carried out to understand and address mental ill health stigma and discrimination in, and across, contexts.[23]

While stigma and discrimination in the workplace have been reported by individuals with mental health issues globally,[1–18] the majority of studies including anti-stigma or discrimination initiatives have focused on a range of mental health diagnoses, or a few specific diagnoses (eg, schizophrenia or substance use disorders), or have been based in HICs.[24] Consequently, there is a gap in research on experienced and/or anticipated stigma and discrimination in the workplace among individuals with major depressive disorder (MDD), particularly in LMICs.[8] This is despite the fact that MDD is recognised to be one of the leading causes of the global burden of disease and is one of the most prevalent causes of disability.[25–27] Furthermore, global evidence suggests that the impact of depression in the workplace is considerable across all countries, regardless of a country's economic development, national income or culture.[28 29]

A global survey conducted by the European Union-funded Anti Stigma Programme European Network (EU-ASPEN) and the International Study of Discrimination and Stigma Outcomes for Depression (INDIGO) among individuals with a clinical diagnosis of MDD in 40 sites across 35 countries has revealed that 62.5% of participants had anticipated and/or experienced discrimination in the workplace. Another noteworthy finding of the study was that participants from countries with a very high HDI reported higher levels of discrimination.[29 30]

Considering the relative paucity of literature focused on the global experiences of stigma and discrimination in the workplace by individuals with MDD and the high levels of anticipated and experienced workplace discrimination among individuals with MDD indicated by the aforementioned global survey, further research is warranted to explore the dynamics and negotiated reality of this important research topic, which is imperative to develop and modify sustained workplace anti-stigma and

discrimination initiatives and strategies. Hence, the qualitative results of this global mixed-methods study aim to provide further insight into how anticipated and experienced discrimination among individuals with MDD are manifested in the workplace, the relationship between anticipated and experienced stigma and discrimination, and how these experiences differ across countries based on their HDI level.

## METHODS

The qualitative data for this global cross-sectional mixed-method study were gathered as part of the combined EU-funded ASPEN and the global INDIGO studies,[30] examining stigma and discrimination among individuals with MDD in 40 sites across 35 countries. This included 18 European ASPEN study countries (Belgium, Bulgaria, England, Finland, France, Germany, Greece, Hungary, Italy, Lithuania, The Netherlands, Portugal, Romania, Scotland, Slovakia, Slovenia, Spain and Turkey) and 17 INDIGO network countries (Australia, Brazil, Canada, Croatia, Czech Republic, Egypt, India, Japan, Malaysia, Morocco, Nigeria, Pakistan, Serbia, Sri Lanka, Taiwan, Tunisia and Venezuela).

The cross-sectional mixed-methods survey was designed to use locally available resources and researchers in order to enable as many LMICs as possible to participate in the study. Participants were recruited by local research staff in each participating country. Gatekeepers (eg, health and social care practitioners, mental health NGOs) identified individuals with a clinical diagnosis of MDD in the last 12 months who attended specialist mental health services (including outpatient or daycare services in public and private sectors). Participants also had to be able to speak and understand the main local language and be aged 18 years or older. Individuals who were receiving psychiatric in-patient care during recruitment were excluded as well as patients with comorbid alcohol and other substance abuse issues. Each study site aimed to recruit a minimum of 25 participants to interview, including young people (18–24), adults (25–65), and older (≥65) people, and twice as many women as men to reflect the increased MDD prevalence in women compared with men.

Written informed and understood consent was obtained from all participants, following a thorough description of the study. Face-to-face interviews were carried out by researchers (who were not involved in participants' care). Data gathering for a combined 40-site ASPEN and INDIGO studies concluded in December 2012. All qualitative framework data analyses from this large global study were carried out thereafter.

The mixed-method cross-sectional survey included the Discrimination and Stigma Scale (version 12; DISC-12), a standardised structured interview that examines experienced and anticipated discrimination among individuals with a mental disorder.[31 32] The scale includes 32 questions with participants being asked to rate the extent to which they have experienced the type of discrimination referred

to in each question on a 4-point Likert scale (0=not at all, 1=a little, 2=moderately, 3=a lot), and open-ended questions to provide examples of their experiences for each question. This qualitative study used the responses from the open-ended components linked to each question of the DISC-12, which are related to employment. However, all comments made in relation to the employment were considered for the final data analysis:

▶ Have you been treated unfairly in finding a job because of your mental health problem? (ie, finding full or part-time work)?
▶ Have you been treated unfairly in keeping a job because of your mental health problem?
▶ Have you stopped yourself from applying for work because of your mental health problem?
▶ Have you been treated more positively in employment because of your mental health problem? (eg, including finding work, keeping work and adjustments in the workplace).

The quantitative data of experienced and anticipated stigma and discrimination in relation to employment, as well as the more positive experiences reported, were presented in a separate publication.[29] The qualitative interview data were analysed using the framework method. This method was selected because its structure enables researchers to compare data across as well as within cases in a systematic way, which facilitates the analysis of large and/or mixed-method datasets. Thus, the rigour and transparency of the method are thought to benefit the quality of the research. Furthermore, it can be used across a range of epistemological approaches.

Following Gale et al,[33] the framework analysis used in this study involved seven stages: (1) Interviews were audio recorded and transcribed verbatim; (2) Researchers read through the interviews and made note of initial thoughts and ideas on the data, familiarising themselves with the material; (3) The third coding stage started with the selection of ten transcripts which researchers then independently openly 'coded' (or highlighted and labelled) text that was considered important using inductive reasoning; (4) Discussion between the researchers regarding their coding, the general similarities, talking through and resolving the differences, led to the development of a provisional analytical framework; (5) Using this framework, one of the researchers coded all of the interviews included in the study using NVivo V.9 software, which necessarily involved refining, revising and creating new codes and categories; (6) Using the coding categories from the coding of the full dataset, data were summarised case by case in an Excel spreadsheet and (7) The final stage of the analysis involved the interpretation of the data, discussing the various codes, categories or themes emerging both between and across participants.

Differences across participants from countries categorised according to the HDI ranking were also explored.[34] This method of categorisation was chosen, as it includes measures of health, education and economic growth, rather than providing a measure of economic growth

alone. Countries were thus categorised as 'very High', 'high', 'medium' or 'low' HDI countries.

In addition to the use of rigorous framework analysis, trustworthiness of the data analysis was further enhanced through triangulation between the literature review, quantitative data and qualitative data. All qualitative data analysis was carried out at the KCL Lead study site.

### Patient and public involvement

Patients and other key stakeholder representatives were consulted at the start of the study to inform the study design and the development and adaptation of the mixed-method interview survey (DISC-12). They were also involved in the piloting of DISC-12, its translation, back-translation and cultural adaptation for the purposes of this study. Key stakeholders, including patient representatives, health and care professionals, academics, charity and NGO representatives, policy-makers and others, have been involved in the dissemination process of findings (eg, in publications, conferences, advocacy activities).

### RESULTS

### Sample characteristics

A total of 141 of the 196 qualitative interviews available were considered to have met the criteria to be included in the framework analysis. Interviews were excluded from the analysis if they (1) contained no examples illustrating the answers participants provided to the DISC-12 items,

(2) if the question was not relevant to them (eg, those who had been retired for a number of years) or (3) if the answers they gave were not apparently relevant to the question asked, that is, in regard to mental health-related stigma and discrimination.

Table 1 shows the characteristics of the participants who were included in the analysis and those excluded, alongside those of the Overall Global Study Sample (n=1087) of participants in the combined ASPEN/INDIGO mixed-method studies. The average age of the included sample was slightly lower than that of the other sample groups, and there were more single people among the included participants. There was a lower proportion of participants with a higher educational level in the included sample relative to the Overall Global Study Sample and expectedly more participants in employment than in the other samples. Perhaps the most significant difference between the samples for this report resides in the HDI groupings, with a lower proportion of participants in the included sample coming from the medium/low HDI group. Half of all those interviewed from the medium/low groups were excluded from the analysis due to the limited information available regarding workplace issues contained within the interviews.

### The framework analysis

The framework analysis was split into six 'frames' (sections) exploring: (1) participants' reported experiences of

**Table 1** Characteristics of participants included in, and excluded from, the framework analysis sample, and those from the overall study sample

| | Included (n=141) n, % | Excluded (n=55) n, % | Total qualitative sample (n=196) n, % | Overall global study sample (n=1087) n, % |
|---|---|---|---|---|
| Gender | | | (missing=6) | |
| Female | 94 (66.7) | 36 (65.5) | 130 (66.3) | 717 (66.0) |
| Age | | | (missing=27) | |
| Mean age | 41.8 years | 45.0 years | 42.7 years | 44.9 years |
| Ethnicity | 11 (7.8) | 3 (5.5) | (missing=44) 14 (7.1) | 70 (6.0) |
| Ethnic minority | | | | |
| Marital status | | | (missing=2) | |
| Married/cohabiting | 65 (46.1) | 31 (56.4) | 96 (49.0) | 542 (50.0) |
| Widowed/separated/divorced | 31 (22.0) | 11 (20.0) | 42 (21.4) | 244 (22.0) |
| Single | 45 (31.9) | 11 (20.0) | 56 (28.6) | 296 (27.0) |
| Qualifications | | | (missing=25) | |
| Up to 18 years | 58 (41.1) | 30 (54.5) | 88 (44.9) | 475 (44.0) |
| Over 18 years | 65 (46.1) | 18 (32.7) | 83 (42.3) | 601 (55.0) |
| Employment status | | | (missing=4) | |
| In employment | 81 (57.4) | 15 (27.3) | 96 (49.0) | 536 (49.3) |
| Unemployed/student/retired | 59 (41.8) | 37 (67.3) | 96 (49.0) | 538 (49.5) |
| Human Development Index | | | | |
| Very high | 75 (53.2) | 21 (38.2) | 96 (49.0) | 503 (46.3) |
| High | 41 (29.1) | 9 (16.4) | 50 (25.5) | 314 (28.9) |
| Medium/low | 25 (17.7) | 25 (45.5) | 50 (25.5) | 270 (24.8) |

employment stigma and discrimination because of their MDD; (2) the reported impact on them of experienced employment stigma and discrimination; (3) the stigma and discrimination they anticipate to experience in the workplace because of their MDD; (4) ways of coping with employment stigma and discrimination; (5) the positive work experiences and related impacts participants described; and lastly, and (6) the reported contextualisation (eg, socioeconomic, economic and cultural aspects) of employment stigma and discrimination. Table 2 illustrates the themes and subthemes identified within each frame of the framework analysis.

As highlighted previously, the quantitative results of the combined ASPEN-INDIGO studies suggested that

**Table 2** Framework analysis of anticipated and experienced stigma and discrimination in the workplace among individuals with major depressive disorder: frames, key themes and subthemes

| Frame | Key themes | Subthemes |
|---|---|---|
| Experiences of employment stigma and discrimination because of mental ill health | Lack support/understanding | Unreasonable work demands |
| | | Sick leave |
| | Abuse and exploitation | Verbal abuse and bullying by colleagues and seniors |
| | | Exploitation by employees |
| | Lack of respect for people and their capabilities ('thwarted progress') | Less competent |
| | | Weaker |
| | | No promotion |
| | Fired/not hired | Lost job |
| | | Not offered employment |
| | Avoided or shunned | Avoided by colleagues |
| | | Avoided by seniors and clients |
| | Lack of confidentiality | Coercion to reveal |
| | | Revelation by others |
| Impacts of experienced employment stigma and discrimination | Exacerbation of mental health problems | |
| | Leaving work or long-term sick leave | |
| | Inability to work | |
| | Reduced confidence in seeking employment | |
| Anticipated employment stigma and discrimination | Concealing | Fear of losing work |
| | | Fear of lack of career progression |
| | | Belief of lack of support |
| | | Belief of lack of understanding |
| | Stop self from applying for work | |
| Coping with employment stigma and discrimination | Avoiding work | |
| | Concealing as coping | |
| | Work harder | |
| | Less stressful employment | |
| | Talking more about depression | |
| Positive employment experiences and impacts | Support, help, understanding | Time off from work |
| | | Reduced workload or pressure at work |
| | Friendships through work | |
| | Sense of achievement/confidence | |
| | Work as a place to escape from difficulties | |
| | Social status of work | |
| Contextualisation of employment stigma and discrimination | Economic climate (locally and globally) | Lost work due to bankrupted employers |
| | | Increased competition for jobs |
| | Sociocultural issues | Family not allowing to work |

**Table 3** Experiences of discrimination by HDI rank

| Key themes | Very high HDI (%) | High HDI (%) | Medium/low HDI (%) |
|---|---|---|---|
| Lack support/understanding (n=30, 21.3%) | 22 (29.3) | 5 (12.2) | 3 (12.0) |
| Abuse and exploitation (n=28, 19.9%) | 17 (22.7) | 6 (14.6) | 5 (20.0) |
| Lack of respect (n=27, 19.1%) | 16 (21.3) | 5 (12.2) | 6 (24.0) |
| Thwarted progress (n=13, 9.2%) | 8 (10.7) | 2 (4.9) | 3 (12.0) |
| Fired/not hired (n=23, 16.3%) | 14 (18.7) | 6 (14.6) | 3 (12.0) |
| Fired (n=18, 12.1%) | 11 (14.7) | 5 (12.2) | 2 (8.0) |
| Not hired (n=10, 7.1%) | 7 (9.3) | 2 (4.9) | 1 (4.0) |
| Avoided or shunned (n=19, 13.5%) | 14 (18.7) | 4 (9.8) | 1 (4.0) |
| Lack of confidentiality (n=10, 7.1%) | 7 (9.3) | 0 (0.0) | 3 (12.0) |

HDI, Human Development Index.

there were relatively high levels of anticipated and experienced discrimination in the workplace. Interestingly, this was significantly higher for those in the very high HDI country group; therefore, the analysis will explicitly focus on the apparent similarities and differences between HDI groupings.[34]

### Frame 1: experiences of employment stigma and discrimination because of mental ill health

Six key themes were identified from the analysis concerning participants' experiences of employment discrimination across HDI groups, as shown in table 3.

### Lack of support and understanding

Around one-fifth of the sample (n=30) reported that they had experienced a lack of support and understanding in the workplace, predominantly from their employers. More than double the proportion of participants in the very high HDI group described difficulties in this area as opposed to those in the other HDI groups. Some of the issues highlighted by the participants centred around a general misunderstanding of the nature of, and recovery from, depression in the workplace. This general misunderstanding and concomitant lack of support appeared to manifest itself in two specific areas:

(1) Unreasonable work demands—participants felt that their employers expected too much of them, and that allowances or accommodations for the impact of depression on their working lives were lacking or not recognised. This was particularly strongly reported in the very high HDI group.

> This is what you signed up to do, you need to get your head together on that son. We can't be having that, you're going to be dealing with worse things than this.—[Very High HDI]

> If I tell my boss that I will not be able to fulfil a special task because the pressure of time is too big he just ignores it. He only says 'This is easy, I have calculated how long it will take and you have to do it.'—[Very High HDI]

(2) Sick leave—mostly these problems came from within the management structure due to either not understanding or believing a person's mental health problems and/or their need to take sick leave, including participants describing being scolded for taking necessary sick leave.

> My employer called me several times when I was on the sick leave to get back to work. What is wrong with you, he said. I do not see that you are ill.—[High HDI]

> When I work I get tired, so I take leaves of absence. And when I am absent I need … I mean my boss scolds me, he can't understand why I was absent, and I can't tell him why I was absent.—[High HDI]

> It is extremely unfair. In the end they (~employer) rejected my sick leave because of it (~depression).—[Medium HDI]

### Abuse and exploitation

Around one-fifth of participants (n=28) reported experiencing abuse and exploitation at work. The subthemes within this major theme largely pertained to (1) verbal abuse and bullying meted out by participants' colleagues and in some instances those in senior positions and (2) exploitation of employees rendered vulnerable by their position as an employee with mental health problems, by those in power.

> I listened to a discussion between two colleagues. I confided in one of them before, for example, that I really felt very bad and had suicidal thoughts. The male colleague just said 'Let the fat pig kill himself, so what?—[Very High HDI]

> Again in a joking manner, people say 'don't mess with that lunatic' behind my back. Because I like them, I am not offended by this. Still it's not nice they say that at work!—[High HDI]

> My regional manager said: 'Yes, but, a fool like you, I don't want to lose'…'to you I may ask what I want',

she said, 'You do not dare to say no'. This was because of my condition.—[Very High HDI]

I was attacked by my employer, when I asked him to give me my salaries that he did not pay regularly. He told me that I am mentally ill.—[High HDI]

### Lack of respect for people and their capabilities: ('thwarted progress')

Again around one-fifth (n=27) of the participants reported that both their employers and colleagues had opined that they were (1) less competent and unable to manage tasks at work when their mental health problems became known. Participants also reported that they felt they were being (2) seen as weaker and less respected by others in the workplace. A number of participants who reported this erosion of respect also said that their opportunities for progress at work were being thwarted (n=13). This included decreased access to career-enhancing training opportunities, being demoted or not considered for internal promotion and being side lined at work. A couple of participants stated that they were (3) not being promoted or moved at work as they wanted because of concerns about the impact of their mental health problems on their productivity and presence, effectively leaving people in employment stasis.

When it suits them to move me it's fine but when I want to go and do something that I want to do, I'm told I need this period of 'stability'.—[Very High HDI]

I knew that my manager wanted to demote me. It was the third time demotion. I was very angry this time. I didn't want to come back to work for company.—[High HDI]

I have been denied promotion because of my illness. They say they can't trust me with higher responsibility.—[Low HDI]

My boss treats me as if I was a kid. She talks to me any how and does not consider me feelings—[Low HDI]

### Fired or not hired

Just under one-fifth of the overall sample (n=23) said that they had either (1) lost their job in some way (n=18) or had (2) not been offered employment due to their mental health problems (n=10). Some of the participants talked about how they were struggling to cope at work or to perform well at a job interview as a result of their depression or the effects of the medication they were taking, and that it was this resultant poor performance that had led to their being fired, or not hired. While this was experienced as reasonable by some, others were less convinced of this. During the recruitment process, some felt that the disclosure of their depression effectively shut down their interviews. Where people had been 'let go', in many cases it was remarked that it was made clear—although unofficially, potentially because of the contravention of employment law—that this was because of their mental health irrespective of the impact it had on their performance.

A couple of participants also talked about having been 'let go' under the guise of employers' concerns for their health, although this was felt to be insincere.

They dismissed me just two days before my probation period was over… my boss knew that he had to employ me at least till the end of the year because of my severe 50% disability. But this is only effective after the probation period.—[Very High HDI]

I had been honest about what I had done the last two years. So yes I had been depressed. And then the conversation was done. Employers fear that the disease comes back… Officially it is something else of course, but they told me clearly that it was due to my illness.—[Very High HDI]

At the end of a job interview, the employer gave me to understand that he wasn't hiring me because he was afraid of the absenteeism that might be caused by my depression.—[Very High HDI]

My manager fired me from my last job because I was praying too much and slept on duty…I was always sleeping because of the side effects of the medicine.—[Low HDI]

### Avoided or shunned

13.5% of participants (n=19) stated that they had been actively avoided or shunned in the workplace, which was again more prevalent in the very high HDI group. Most of those reporting being shunned described (1) being avoided at work by their colleagues who were aware of their mental health problems. This was either done by excluding people from conversations or activities, not contacting them when they were on sick leave, or just by the perception of a distance forming. There were also slightly less frequent reports of people (2) being avoided by senior staff, their customers or clients. Interestingly, individuals in the very high HDI group described this avoidance twice as much as the other HDI groups.

I am a manager. In the beginning, they used to call me, but then it was as if some kind of wall has raised. When they found out that I was on a psychiatric ward, they wrote me off saying I was a lunatic.—[High HDI]

He wouldn't phone me about work related things because he didn't know the reaction he would get or what to say to me so he would work around it which would have an affect a little bit on the business.—[Very High HDI]

Right now even, at work I have no idea what my future is and the people that are responsible for that are nowhere in sight.—[Very High HDI]

Since I'm sick and I no longer work, people (~friends) ask after me less—[Medium HDI]

### Lack of confidentiality

10 participants reported having their confidentiality not respected in the workplace. Of these, some were expressing their unhappiness at (1) being coerced into

revealing their mental health status in their employment, for others it was more to do with (2) having confided in colleagues who went on to discuss their mental health problems with others. One participant talked of how reasons for sick leave were not confidential and were widely circulating.

> I just told my friends. And then they talked to each other, now there may be one to two hundred people knew.—[Medium HDI]

> When I came back to my company from the hospital, the human resources wanted to know my diagnosis, I don't know if this is relevant to this question or not. I just didn't tell them, I said that was my personal business.—[Very High HDI]

### Frame 2: impacts of experienced employment stigma and discrimination

Although this was not an explicit question asked of participants, some of the participants described how their experiences of discrimination had impacted on them. Some participants suggested that this had exacerbated their mental health problems (n=8), with people feeling ashamed, humiliated, angry and upset by events encountered in their workplace. The negative experiences also appeared to lead a number of participants to either contemplate leaving or actually leave their jobs (including taking early retirement where possible) or to take long-term sick leave (n=5), which further eroded their confidence in seeking employment (n=11). Over one-third of the participants reported that they felt unable to work or look for work because they were concerned that they were still struggling with their mental health, felt unable to control their depression in order to work or that the depression itself had reduced their confidence in seeking employment. More participants in the very high and high HDI groups stated this. Interestingly, the majority of the participants who expressed this view had also actually experienced discrimination in the workplace. It is, therefore, possible that this erosion of confidence is related to discrimination and not just mental health symptoms.

> …after ten or fifteen years of being self-employed thinking I could work for somebody else, that one experience was one of the most humiliating I guess I've gone through in a long time and I kind of decided I wouldn't apply for any work.—[Very High HDI]

> I stopped work. Yes, I have, but again, it is more by people's actions and what they have been sort of

saying and their attitude, for want of a better word, towards me.—[Very High HDI]

### Frame 3: anticipated employment stigma and discrimination

Overall, 60% of the included sample felt that they would be subject to discrimination and stigma for their mental health problems in the workplace which led to individuals either (1) stopping themselves from seeking employment or (2) concealing their mental health problems from employers. Both of these were more prevalent in the very high HDI group than the other groups as shown in table 4.

#### Concealing depression from current and prospective employers

Almost 40% of the overall included sample reported concealing their mental health status at work, which was more prevalent in the very high HDI countries group. Aside from simply not talking about their difficulties, people also tried to present a happier façade within the workplace or they preferred to say that they had problems with their physical health or tiredness. More than three-quarters of the participants who reported concealing their mental health status had not described any previous experiences of discrimination. Those who did experience discrimination, most frequently reported a lack of support and understanding (n=15) and thwarted progress (n=8). Overall, the main reasons why participants described concealing their depression was because of (1) fear that they will lose their job or not get hired in the first place if they reveal their mental health status (n=12), (2) fear that it will damage their career or thwart their progression at work (n=5) and (3) belief that they will not be supported or that their depression will not be understood (n=5).

> I never show myself in my workplace like this. I am like an actor. As an actor changes for films, I change when I go into work. It's exhausting.—[Very High HDI]

> Perhaps I am stigmatizing myself, but if I should look for a job, I would not tell that I am taking pills. Not for me, I do not care about telling it however perhaps if they would know that you have depression they may push you aside… Perhaps it does not happen, but they reject you because of the pregnancy, because of everything, so….—[Very High HDI]

> The reason I didn't [~disclose] was because then you'll be stigmatised and then when the next pay off comes, 'oh we don't want him because there's a good

| Table 4 | Anticipated discrimination by HDI rank | | |
|---|---|---|---|
| **Key theme** | **Very high HDI (%)** | **High HDI (%)** | **Medium/low HDI (%)** |
| Concealing (n=57, 40.4%) | 36 (48.0) | 14 (34.1) | 7 (28.0) |
| Stop self from applying (n=52, 36.9%) | 35 (46.7) | 12 (29.3) | 5 (20.0) |
| HDI, Human Development Index. | | | |

chance he'll be off for a couple of weeks next year.—[Very High HDI]

…people don't have to know it. This is like haemorrhoids, one doesn't announce it out loud.—[Very High HDI]

### People stopping themselves from pursuing work

Three-quarters of the participants who were not looking for work were those who had also described discriminatory experiences in the workplace (n=40). Around 70% of those who had been 'fired' (n=13), not hired (n=7) or had their progress thwarted at work (n=9), stopped themselves from pursuing work. Around half of those who had been 'abused' (n=16), avoided (n=16), not supported (n=14) or respected (n=14) also reported that they had stopped looking for work. Nearly half of those in the very high HDI country group were not pursuing employment, which is a far higher proportion than any of the other HDI groups.

Maybe they were right, 'what will a person with a disease like this do for a business?'…—[Very High HDI]

…when I go to look for a job, for example: 'Are you sick, or not?', 'What do you have exactly?', 'Are you a patient?', 'Do you have whatever…'. You understand? I tell them I am sick, I have to tell them because if they notice … they say they are not interested. Because of this, I refrained from applying.—[Medium HDI]

### Frame 4: coping with employment stigma and discrimination

Reported 'coping' here relates to both positive and negative coping strategies to deal with experienced and/or anticipated employment stigma and discrimination. Accordingly, (1) 'avoiding applying for work' or (2) 'concealing a mental health problem in the workplace' could therefore be seen as a means of coping with the experience and impact of employment discrimination. Indeed, these were by far the most reported practised means of coping. There were some alternative means of coping with workplace discrimination identified in the analysis reported by a small number of individuals, and largely centring around people reporting that they (3) worked harder to prove they were capable of the work or to try to get back into a normal work pattern following a period of absence. For some, this worked well, for a few it left them more tired and less able to cope in the end. A couple of participants also talked about (4) applying for less stressful types of employment, or that they actually (5)

talked more about their depression and found common ground with others in the workplace.

In the time of my internship, I was really thinking like, 'you must work hard, you gotta work hard, or else you don't exist'.—[Very High HDI]

When I am feeling worst I just call in sick and so not go, putting any silly excuse as having a headache for example.—[Very High HDI].

I am tired. I can't be productive, as I am tired and sad, but I don't reveal that to my friends, I resist and reach their productivity.—[Medium HDI]

…we meet we have a great talk. Sometimes they pour out their heart. They know about my previous illness. I can understand their problems better. I can help them because of the former experience. They ask me how I managed to overcome it…—[Very High HDI]

### Frame 5: positive employment experiences and impacts

It is also very important to highlight that just under 40% of the sample (n=56) reported having positive experiences and related impacts at work, with almost half (n=12) of the medium/low HDI group included in the analysis reporting non-discriminatory behaviour. This was higher than for the very high (n=28) and high (n=16) HDI groups.

### (1) Support, help, understanding

In general, the positive experiences described included being supported and helped, being cared about/for and being encouraged by employers and colleagues. However, there were two specific areas that participants predominantly reported on, as shown in table 5 and highlighted below.

(1) Time off from work: Around 10% of the sample stated that they had been given time off work because of their depression without subsequently experiencing problems because of it. Mostly, time off was given for either treatment and/or for longer periods of time in order help people recover. The descriptions given suggested that employers appeared to be genuinely concerned and supportive of people, rather than the time off being seen as a black mark against the person, although a few people observed that this was also in accordance with workplace rules and legislation.

…you ask for 2 or 3 weeks off, it's not something the company can really deal with, but they consider my

**Table 5** Positive and non-discriminatory experiences by HDI rank

| Key theme | Very high HDI (%) | High HDI (%) | Medium/low HDI (%) |
|---|---|---|---|
| Support, help, understanding (n=56, 39.7%) | 28 (37.3) | 16 (39.0) | 12 (48.0) |
| Time off (n=14, 10.6%) | 7 (9.3) | 4 (12.2) | 3 (12.0) |
| Reduced workload/pressure (n=16, 12.0%) | 9 (12) | 3 (7.3) | 4 (20.0) |

HDI, Human Development Index.

condition, saying "take care of your health, first," and including the fact that they allow me to put priority on recuperating…—[Very High HDI]

…my boss has been very supportive. I had to take 3 months off during my initial state of illness, and he just asked me to rest well at home and then only return to work…—[Medium HDI]

…my principal has been generous in understanding my illness, giving me off days and extending my job deadlines…—[Low HDI]

(2) Reduced workload or pressure at work: A similar number of participants (10%) reported that their employers had also either reduced their workload, changed the nature of the work they did or otherwise were applying less pressure at work. One participant described feeling that their employer was doing everything possible to help them remain at work.

That's quite a stressful area of work and I said that I couldn't do that, it's really like that there - you feel pressure that something could go wrong. So they don't assign me there…—[Very High HDI]

The office manager knows that I'm sick. So he gives me less work.—[Medium HDI]

During the periods of depression…at work they can tolerate more mistakes or tolerate less discipline because of the depression.—[Medium HDI]

Around 10% of the participants reported (2) making and developing friendships through work as being one of the most positive impacts of working. A number of other positive impacts were also described by a few participants, which included (3) feeling a sense of achievement and confidence through their work, (4) work as a place to escape from difficulties and (5) work giving people a status that alters how they are perceived by others.

…because I finished the university and because I am working people think of you differently, more positively…—[Medium HDI]

At first I had some difficulties, I made some mistakes in my job, because I lost my concentration. My employers did not change my tasks. It was a help for me, they were faithful for me, it was a stimulus for me, very important for me.—[Very High HDI]

### Frame 6: contextualisation of employment stigma and discrimination

Some of the participants commented on the prevailing context within which they were situated in terms of employment. In some ways, there were commonalities between all three HDI groups, particularly around the difficulties that people were generally having in finding work—irrespective of mental health issues—and the pressures and strains this brings with it. A few people had (1) lost work due to their employers becoming bankrupt in all three HDI groups. Two participants suggested that the difficult local and global economic situation and the

increased competition for jobs lead to mental health problems being a further disadvantage. This may be especially pertinent for those over the age of 50—although there were a couple of reports about the problems facing younger people too.

I could have been job-hunting, but I've also kind of thought that given that I have been unemployed for ages, and my diagnosis and things …and the level of unemployment, you know, they're going to get dozens of people applying for every job.—[Very High HDI]

…my firm is closed and now I am unemployed, I work sometimes part time but it is hard to find a job being old as I am…—[High HDI]

In the medium HDI group only, four female participants talked about (2) their family not allowing them to work—although it was unclear whether this was solely gender or mental health related, or a mix of gender, cultural and mental health-related issues at play.

I applied for a very good job, my own family cursed me …I was offered a very good school job but they didn't give me permission to join saying that it is far away and is less paid job …I was doing a job as an educational officer. When I used to come back home from job, faces of all people at home used to be furious. They didn't even like to talk to me.—[Medium HDI]

## DISCUSSION

This paper presents the findings from the qualitative framework analysis undertaken in a subsample of participants with MDD from the combined ASPEN and INDIGO-Depression studies including 40 sites across 35 countries across low-HDI, medium- and high-HDI countries. Hence, the learnings from this global study strengthen the scant evidence pertaining to workplace stigma and discrimination in relation to people with MDD in a global sample.

The present study highlights that, consistent with previous research among individuals with other mental diagnoses, people with MDD both anticipated and experienced stigma and discrimination in the workplace due to their mental health status.[5 35] The ubiquity of these experiences is also evident in a survey of mental health consumers, among whom 36% reported experiencing stigma from their coworkers, colleagues or classmates, and 24% reported stigma from their employers or supervisors.[35] Furthermore, the consequences of anticipated and experienced stigma and discrimination are also consistent with previous research, as in the aforementioned survey, 21% stated that they became less likely to make an application for a job or school because of their experiences of stigma.[35]

Previous research suggests that the anticipation may be a precursor or consequence of experienced discrimination

or both, among people with mental health problems,[31 36] which possibly explains the observed association of anticipated and experienced workplace stigma and discrimination among individuals with MDD in this study.

There has been little insight into whether this group experienced limited support or understanding or were shunned or avoided in the workplace due to their mental health problem. Furthermore, this research demonstrates that, just as for mental illnesses such as schizophrenia or substance use disorders, individuals with MDD also experience significant barriers to pursuing work due to anticipated stigma or discrimination in relation to their mental ill health.[1–18 24–29] In a similar vein, a study conducted in Serbia revealed that people with MDD experienced significant discrimination in keeping a job.[37]

Considering the observed differences across countries based on their HDI level, it is evident that participants from the very high HDI group reported higher levels of anticipated and experienced discrimination than the other two groups, specifically feeling less support and understanding and being more avoided/shunned in the workplace, and in people stopping themselves from looking for work because of fears of discrimination. Furthermore, participants from the medium/low HDI were more likely to report positive experiences in the workplace. This corroborates findings from other studies.[19–29]

A similar trend was observed in the ASPEN-INDIGO global study involving 834 people with MDD, in which significantly higher percentages of workplace discrimination was reported in very high HDI countries in comparison to medium/low HDI countries.[29] The highly competitive workplaces and highly individualised societies with limited family and/or community support[19–23 38–41] prevailing in the very high HDI countries as opposed to medium/low HDI countries could be considered as possible reasons for this observed trend of discrimination and stigma in countries across HDI level. In high/very high HDI countries, the process of returning to employment following a period of mental ill health can be very challenging in an environment with workplace stigma and poor employee mental health support.[41]

In contrast, in low-income countries where the economy is often more informal, people with a history of mental ill health are likely to be found roles within the workplace which are easier to resume after a period of illness.[42–44] In addition, in high/very high HDI countries, we find that the work environment is typically impersonal and competitive so that even when a person recovering from a severe episode of major depression finds a job, the sense of exclusion and marginalisation continues.[21 43]

Even though the global literature on the association between perceived stigma and education level in people with MDD fails to provide conclusive evidence, some studies suggest a positive association between higher education and greater perceived stigma[39] and discrimination.[40] In this context, it could also be assumed that the concept of perceived stigma and discrimination may not

be widely known and hence resulting in low reporting in countries with low HDI level.

That said, this qualitative study suggests that workplace stigma and discrimination of people with MDD continues to be a very serious and commonly experienced problem across many countries—regardless of HDI—which is an important finding. It adds additional insights and explanation to the previous quantitative counterpart study.[29] The two studies are exceptional in methodology, involving 35 different countries and cultures, which makes them quite unique. Despite similarities, both studies differ and so do their findings. The previous quantitative study paper, assessed whether people with MDD: (1) anticipated and experienced discrimination when trying to find and keep paid employment and (2) if participants in high, middle and lower developed countries differed in these respects, and if discrimination experiences were related to employment status. It showed that 63% of respondents had anticipated and/or experienced discrimination in the work setting, and that having experienced workplace discrimination was independently related to unemployment.[29] By contrast, the current qualitative study gives more detailed and in-depth information showing, for instance, that people with MDD often encounter a lack of support and understanding, experience abuse, exploitation and a lack of respect, are more often fired or not hired, avoided or shunned. Importantly, this study also shows that these discriminatory experiences negatively affected people's sustainable employability. For instance, they often had stopped themselves from pursuing work. This underlines the negative impact of stigma on health and well-being because unemployment is related to poverty, ill mental health and a higher risk of suicide. This study also found that countries with a higher HDI reported higher stigma.

In the past 10–15 years, there have been some significant changes worldwide—including also in the economy—with staff shortages in many countries and more workplace flexibility, which are factors that are beneficial for work participation of people with mental illness.[45] The topic of workplace stigma has received increasing attention over the past two decades but, by far, most studies have been conducted in North America and Europe.[13] The findings of several recent studies—including this present study—uggest that, in general, workplace mental ill health stigma remains a highly understudied and underestimated topic despite its severe consequences on peoples' overall well-being, sustainability of employment and income (and linked poverty).[2–4 46 47] The current study findings add to this evidence, illustrating the detrimental effects on people and indicate the need for much more applied research and action, in this respect, especially also in LMICs.

## Strengths and limitations

The findings of this global study are significant because they point to the need to both promptly recognise and effectively address stigma and discrimination in the workplace for individuals with MDD, and that these

experiences are not exclusive to individuals with mental illnesses such as schizophrenia or substance use disorders, for instance.[41] Furthermore, the findings point to the social, economic and potential mental health consequences of such anticipated and experienced discrimination in this population.

The findings of this study that experiences of discrimination in the workplace due to MDD vary across HDI level are also innovative and significant. The ASPEN-INDIGO study is thus far the first one and this present study is the first qualitative framework analysis to examine whether the human development level is associated with experienced or anticipated stigma and discrimination and demonstrates that individuals from very high HDI countries are more likely to report both anticipated and experienced discrimination in the workplace. Further research is needed to investigate why these differences exist, and strategies to appropriately address these experiences globally as well as in the local context.

This study also has several limitations. First, approximately one-fourth of the interviews available were not included due to limited information regarding workplace issues contained within the interviews. Second, as almost half of the available interviews with participants from medium/low HDI groups were excluded, there is also the potential that the experiences of this group are not sufficiently represented. In addition, it may be that people with MDD are subject to observer-expectancy effect leading to information bias resulting in possible under-reporting or over-reporting of stigma and discrimination. Third, while having a large study sample from 35 countries providing useful insights across different HDI countries is a main strength of the study, owing to the relatively smaller number of participants from each included country, these findings should be viewed as exploratory with more in-depth local research needed. Finally, the study's methodology (through mixed-methods survey tool) did not sufficiently allow to give a solid answer as to why individuals from very high HDI countries were likely to report more anticipated and experienced workplace discrimination because we used the open answers in the questionnaires. Therefore, the methodology did not allow to seek for data saturation in the way qualitative interviews would normally do, which allows them to explore this in more detail. Future studies will have to address this further, both within and across different contexts and workplaces.

## CONCLUSIONS

This study provides important insights into experiences of stigma and discrimination in the workplace among individuals with MDD across 40 sites in 35 countries. The study represents the first such global research examining the experiences of this group, filling an important gap in research on stigma and discrimination surrounding MDD. Furthermore, the research illuminates important relationships that may exist between country of origin and stigma and discrimination, identifying that individuals from very high HDI countries were more likely to report anticipated and experienced discrimination in the workplace.

The findings have important implications for initiatives to address stigma and discrimination related to mental health, and in particular, point to the need for antistigma campaigns and policy to reduce stigma and discrimination in the workplace among individuals with MDD, both because the stigma and discrimination experienced by this group has been previously overlooked, and ultimately, because of its implications for workforce participation and mentally healthy workplaces. Further research is needed to provide more insight into the relationship between HDI level and discrimination, more understanding of stigma and discrimination within and across contexts, context-appropriately tailored strategies to address such stigma and discrimination, and additionally, how best to address stigma and discrimination experienced in various workplaces worldwide among individuals with other mental illnesses (other than MDD).

**Author affiliations**
[1]Leicester School of Allied Health Sciences, De Montfort University, Leicester, UK
[2]Cambridge Public Health, Department of Psychiatry, University of Cambridge, Cambridge, UK
[3]Department of Community Medicine, Rajarata University of Sri Lanka Faculty of Medicine and Allied Sciences, Saliyapura, Sri Lanka
[4]Department of Criminology, Swansea University, Swansea, UK
[5]Rashid Latif Khan University, Lahore, Pakistan
[6]Faculty of Medicine, University of Tunis El Manar, Tunis, Tunisia
[7]Research Institute for Primary Care & Health Sciences, School of Medicine, Keele University, Keele, UK
[8]Institute for Global Health, Faculty of Health, Keele University, Staffordshire, UK
[9]Faculty of Medicine, University of Ljubljana, Ljubljana, Slovenia
[10]Faculty of Medicine, Ain Shams University, Cairo, Egypt
[11]Psychiatric Center, Ibn Rushd University, Casablanca, Morocco
[12]Associacao para o Estudo e Integracao Psicossocial, Lisbon, Portugal
[13]Mental Health Foundation, London, UK
[14]University of Strathclyde, Glasgow, UK
[15]LUCAS, University Hospitals KU Leuven, Leuven, Belgium
[16]Department of Neuroscience, Biomedicine and Movement Sciences, University of Verona, Verona, Italy
[17]Health Service and Population Research, Institute of Psychiatry, Psychology and Neuroscience, King's College London, London, UK
[18]Tranzo Scientific Center for Care and Welfare, Tilburg School of Social and Behavioral Sciences, Tilburg University, Tilburg, Netherlands
[19]Phrenos Centre of Expertise, Utrecht, Netherlands
[20]Tilburg School of Social and Behavioral Sciences, Tilburg University, Tilburg, The Netherlands

**Collaborators** ASPEN-INDIGO Study Group Credit Statement The ASPEN-INDIGO staff at coordinating centres (at the time the studies took place): Graham Thornicroft, Tine Van Bortel, Samantha Treacy, Elaine Brohan, Shuntaro Ando, Diana Rose (King's College London, Institute of Psychiatry, London, England); Kristian Wahlbeck, Esa Aromaa, Johanna Nordmyr, Fredrica Nyqvist, Carolina Herberts (National Institute for Health and Welfare, Vaasa, Finland); Oliver Lewis, Jasna Russo, Dorottya Karsay, Rea Maglajlic (Mental Disability Advocacy Centre, Budapest, Hungary); Antonio Lasalvia, Silvia Zoppei, Doriana Cristofalo, Chiara Bonetto (Department of Public Health and Community Medicine, Section of Psychiatry, University of Verona, Italy); Isabella Goldie, Lee Knifton, Neil Quinn (Mental Health Foundation, Glasgow, Scotland); Norman Sartorius (Association for the improvement of mental health programmes (AMH), Geneva, Switzerland). The ASPEN/INDIGO

staff at partner centres: Chantal Van Audenhove, Gert Scheerder, Else Tambuyzer (Katholieke Universiteit Leuven, Belgium); Valentina Hristakeva, Dimitar Germanov (Global Initiative on Psychiatry Sofia, Bulgaria); Jean Luc Roelandt, Simon Vasseur Bacle, Nicolas Daumerie, Aude Caria (Etablissement Public Sante´ Mentale Lille-Métropole (EPSM/CCOMS), France); Harald Zaske, Wolfgang Gaebel (Heinrich-Heine Universitat Dusseldorf, Rheinische Kliniken Dusseldorf, Germany); Marina Economou, Eleni Louki, Lily Peppou, Klio Geroulanou (University Mental Health Institute (UMHRI (EPIPSI), Greece); Judit Harangozo, Julia Sebes, Gabor Csukly (Awakenings Foundation, Hungary); Giuseppe Rossi, Mariangela Lanfredi, Laura Pedrini (IRCCS Istituto Centro San Giovanni di Dio Fatebenefratelli, Brescia, Italy; Arunas Germanavicius, Natalja Markovskaja, Vytis Valantinas (Vilnius University, Lithuania); Jaap van Weeghel, Jenny Boumans, Eleonoor Willemsen, Annette Plooy (Stichting Kenniscentrum Phrenos (KcP), The Netherlands); Teresa Duarte, Fatima Jorge Monteiro (Associacao para o Estudo e Integracao Psicossocial, Portugal); Radu Teodorescu, Iuliana Radu, Elena Pana (Asociatia din Romania de Psihiatrie Comunitara, Romania; Janka Hurova, Dita Leczova (Association for Mental Health INTEGRA, o. z., Slovakia); Vesna Svab, Nina Konecnik (University Psychiatric Hospital, Slovenia); Blanca Reneses, Juan J Lopez-Ibor, Nerea Palomares, Camila Bayon (Instituto de Psiquiatria at the Hospital Universitario San Carlos, Spain); Alp Uçok, Gulsah Karaday (Foundation of Psychiatry Clinic of Medical Faculty of Istanbul (PAP), Turkey); Nicholas Glozier, Nicole Cockayne (Brain Luís Fernando Tófoli, Maria Suely Alves Costa (Universidade Federal do Ceara´, Campus Sobral, Brazil); Roumen Milev, Teresa Garrah, Liane Tackaberry, Heather Stuart (Department of Psychiatry, Queen's University, Canada/Providence Care, Mental Health Services, Kingston, Ontario, Canada; Branka Aukst Margetic, Petra Folnegovic Groiæ (Department of Psychiatry, University Hospital Centre ZagrebMiro Jakovljeviæ, Croatia); Barbora Wenigova´, elepova´ Pavla (Centre for Mental Health Care Development, Prague, Czech Republic); Doaa Nader Radwan (Institute of Psychiatry, Ain Shams University, Cairo, Egypt); Pradeep Johnson, Ramakrishna Goud, Nandesh, Geetha Jayaram (St. John's Medical College Hospital, St John's National Academy of Health Sciences, Bangalore, India; Shuntaro Ando (Social Psychiatry, Tokyo Metropolitan Institute of Medical Science, Tokyo, Japan; Yuriko Suzuki, Tsuyoshi Akiyama, Asami Matsunaga, Peter Bernick (NTT Kanto Hospital, Japan); Bawo James (Federal Neuropsychiatric Hospital, USELU, Benin City, Nigeria; Bolanle Ola, Olugbenga Owoeye (Federal Neuropsychiatric Hospital Yaba, Lagos, Nigeria); Yewande Oshodi (Department of Psychiatry, College of Medicine University of Lagos and Lagos University Teaching Hospital, Lagos, Nigeria; Jibril Abdulmalik (Federal Neuropsychiatric Hospital, Maiduguri, Nigeria); Kok-Yoon Chee, Norhayati Ali (Kuala Lumpur Hospital and Selayang Hospital, Malaysia); Nadia Kadri, Dounia Belghazi, Yassine Anwar (Ibn Rushd University Psychiatric Centre, Casablanca, Morocco); Nashi Khan, Rukhsana Kausar (University of the Punjab, Department of Applied Psychology and Centre for Clinical Psychology, Lahore, Pakistan); Ivona Milacic Vidojevic (Faculty for Special Education and Rehabilitation, Belgrade, Serbia); Athula Sumathipala (Institute of Psychiatry, King's College London/Institute for Research and Development, Sri Lanka); Chih-Cheng Chang (Chi Mei Medical Centre, Department of Psychiatry, Tainan), Taiwan; Fethi Nacef, Uta Ouali, Hayet Ouertani, Rabaa Jomli, Abdelhafidh Ouertani, Khadija Kaaniche (Razi Hospital Manouba, Department of Psychiatry, Tunis, Tunisia); Ricardo Bello, Manuel Ortega, Arturo Melone, María Andréina, Francisco Marco, Arturo Ríos, Ernesto Rodríguez, Arianna Laguado (Hospital Universitario de Caracas, Caracas, Venezuela).

**Contributors** TVB was the overall ASPEN-INDIGO Scientific Programme Coordinator and a Senior Researcher on the studies. GT was the principal investigator and guarantor of the overall ASPEN-INDIGO study and conceptualised the overall mixed-method cross-sectional DISC-12 survey study. This qualitative substudy was planned and conceptualised by TVB, NDW and ST, based on the availability of open-ended answers from the mixed-method cross-sectional DISC-12 survey data. Coauthors (TVB, ST, NK, UO, AS, VS, DN, NK, MFM, LK, NQ, CVA, AL, CB and JvW) have been involved in local data acquisition. TVB, NDW and ST carried out the qualitative framework analysis and wrote the initial draft manuscript. EB, JvW and NQ provided further significant input. Co-authors (TVB, NDW, ST, NK, UO, AS, VS, DN, NK, MFM, LK, NQ, CVA, AL, CB, GT, JvW and EB) have been involved in the interpretation of findings. TVB, EB, NDW, JvW, CVA, AL and CB carried out the manuscript revisions. All further coauthors (GT, ST, NK, UO, AS, VS, DN, NK, MFM, LK and NQ) were then invited to read, comment on, edit and approve the fully revised manuscript and all changes made.

**Funding** This study arises in part from the Anti Stigma European Network (ASPEN) programme which was in part funded by the European Union Public Health Programme.

**Disclaimer** The funder was not involved in the study design, nor the collection, analysis, interpretation and write up of the data and manuscript.

**Competing interests** None declared.

**Patient and public involvement** Patients and/or the public were involved in the design, or conduct, or reporting, or dissemination plans of this research. Refer to the Methods section for further details.

**Patient consent for publication** Consent obtained directly from patient(s).

**Ethics approval** This study involves human participants and the overall research study was approved by the appropriate ethics review board in the Lead site (main coordinating and data handling site) at King's College London (KCL), UK (Ref. R&D2010/108) and also at the University of Verona (UNIVER), Italy (Ref. CE1729/2009) where, respectively, all qualitative and quantitative secure data handling, data analysis and data storage were carried out. Subsequently, all other data collection sites submitted the KCL Lead site's overall study ethics approval to their respective local clinical out-patient settings for full scrutiny and approval to carry out the local data gathering through DISC-12 surveys. Sites that did not have any ethics approval mechanism or authority in place (which was most sites at that time), followed the KCL Lead/main coordinating site's global mental health study ethics and conduct. No data were stored or handled for analysis in the data collection sites. Analyses (qualitative and quantitative) were carried out by KCL and UNIVER respectively with all local study collaborators consulted on interpretation of findings. A full list of organisations (with corresponding local P.I.s) where this study and KCL Lead site ethics approval was approved and/or followed for data collection, is included in appendices. All participants provided full written informed and understood consent prior to their involvement in the study (all participants were above the age of 18 years and had full capacity to consent).

**Provenance and peer review** Not commissioned; externally peer reviewed.

**Data availability statement** Data are available on reasonable request. The transcribed, translated (in English), fully anonymised and digested data in NVivo, used during this present study, are available from the study principal Investigator on reasonable request. No additional data are available.

**ORCID iDs**
Tine Van Bortel http://orcid.org/0000-0003-0467-6393
Nuwan Darshana Wickramasinghe http://orcid.org/0000-0001-6025-6022

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
