## [Reviewer comments · BMJ Open]

ARTICLE DETAILS

TITLE (PROVISIONAL)	Anticipated and experienced stigma and discrimination in the workplace among individuals with Major Depressive Disorder in 35 countries: qualitative framework analysis of a mixed-method cross-sectional study
AUTHORS	Van Bortel, Tine; Wickramasinghe, Nuwan; Treacy, Samantha; Khan, Nashi; Ouali, Uta; Sumathipala, Athula; Svab, Vesna; Nader, Doaa; Kadri, Nadia; Monteiro, Maria Fatima; Knifton, Lee; Quinn, Neil; Van Audenhove, Chantal Van Audenhove; Lasalvia, Antonio; Bonetto, Chiara; Thornicroft, Graham; van Weeghel, Jaap; Brouwers, Evelien

VERSION 1 – REVIEW

REVIEWER	Kristy Sanderson University of East Anglia Faculty of Medicine and Health Sciences, Health Sciences
REVIEW RETURNED	07-Sep-2023

GENERAL COMMENTS	This paper reports analysis of qualitative (free text) findings from a cross-sectional survey on workplace stigma. The quantitative results were published in 2016. The surveys were conducted in 2012. Main strength of the study is the breadth - 35 countries. Major comment The authors make the most of the free text data with an appropriate and informative analysis. My main comment refers to the limitations of this as qualitative data and how much this adds to the 2016 paper. The conclusion seems to be similar - that stigma is possibly more impactful in higher HDI countries - but the paper does not offer much by way of explanation from the qualitative data as to why this may be the case. There was no/little literature reviewed in the Introduction to help position the question of whether and why the HDI would be related to work stigma and work outcomes for people with depression. There were some brief comments in the Discussion. Given this finding has already been published for the quant data, I feel there needs to be more drawn out from the qualitative to make this a stand-alone paper. The authors are clear on the limitations of missing data from lower HDI countries (half missing data from low/medium HDI countries), but it nonetheless remains a concern that there are few quotes from these countries. Minor comments There has been substantial international effort to address issues around depression in the workplace in the past decade. Can the authors comment on whether they think anything has changed in relation to stigma and employment outcomes in the 10 years since this data was collected - would the results still hold if this study was
--

	conducted now? The free text responses are attached to specific items of the DISC-12, where the items refer to "mental health problem." Were the questions framed to be answered in relation to MDD only, or inclusive of any other mental health condition the participants might have had? This is important to clarify as the authors mention a number of times about this data is novel for MDD in comparison to other disorders including alcohol disorders which are highly comorbid with MDD. Table 1: are there any other employment-related variables from the quantitative survey that could be included to help understand contextual factors of the qualitative data?
--	--

VERSION 1 – AUTHOR RESPONSE

Reviewer Requests:

Comments to the Author:

This paper reports analysis of qualitative (free text) findings from a cross-sectional survey on workplace stigma. The quantitative results were published in 2016. The surveys were conducted in 2012. Main strength of the study is the breadth - 35 countries.

Major comment

[Please note: we have divided the ‘major comment’ into three sections, each with their own tailored response]

1. The authors make the most of the free text data with an appropriate and informative analysis. My main comment refers to the limitations of this as qualitative data and how much this adds to the 2016 paper. The conclusion seems to be similar - that stigma is possibly more impactful in higher HDI countries - but the paper does not offer much by way of explanation from the qualitative data as to why this may be the case.

Response: Indeed, both the 2016 and the current paper find that employment stigma is possibly more impactful in higher HDI countries. They also both suggest that workplace stigma is a very serious and commonly experienced problem across many countries - regardless of HDI - which is an important finding. The two studies are exceptional in methodology, involving so many different countries and cultures, which makes them quite unique. **However, despite similarities, the studies do differ, and so do their findings.** The 2016 paper was a quantitative study, assessing if people with MDD (1) anticipated and experienced discrimination when trying to find and keep paid employment; (2) if participants in high, middle and lower developed countries differed in these respects, and if discrimination experiences were related to employment status. It showed that 63% of respondents had anticipated and/or experienced discrimination in the work setting, and that having experienced workplace discrimination was independently related to unemployment. In contrast, the current study is a qualitative study which gives more detailed and in-depth information, showing for instance that people with MDD often encounter a lack of support and understanding, experience abuse, exploitation and a lack of respect, are more often fired or not hired, avoided or shunned. Importantly, the study also shows that these discriminatory experiences negatively affected people’s sustainable employability. For instance, they often had stopped themselves from pursuing work. This underlines the negative impact of stigma on health and wellbeing, because unemployment is related to poverty, ill mental health and a higher risk of suicide. In the current study, it was found that higher stigma was found in countries with a higher HDI. The study’s methodology did not allow to give a solid answer as to *why* this was the case, because we used the open answers in the questionnaires. Therefore, the methodology did not allow to seek for data saturation in the way qualitative interviews do, and which allows them to go explore this in more detail. Future studies will have to address this further. We have now added this limitation in the ‘Strengths and limitations’ section (Pages 36-37, Lines 774-805). We have also incorporated the abovementioned differences with/ and additional insights to our 2016 paper (Page 34-36, Lines 741-772). Furthermore, we have added more literature and references (see response below).

2. There was no/little literature reviewed in the Introduction to help position the question of whether and why the HDI would be related to work stigma and work outcomes for people with depression. There were some brief comments in the Discussion.

Response: We have now added more literature to the introduction and discussion on the role that socio-cultural factors and cross-national variations play in stigma as well as our previous paper on cross-national variations. However literature on cross-cultural differences in mental ill-health stigma are still very scarce, and even more so in relation to the workplace (almost none existent). Therefore, this paper makes a contribution – based on data from 35 countries – towards this gap in the literature. (Pages 6-8, Lines 119-159 + additional references added)

3. Given this finding has already been published for the quant data, I feel there needs to be more drawn out from the qualitative to make this a stand-alone paper. The authors are clear on the limitations of missing data from lower HDI countries (half missing data from low/medium HDI countries), but it nonetheless remains a concern that there are few quotes from these countries.

Response: More quotes from the qualitative data have been added throughout the Results section, with special attention for the Low and Medium HDI countries where possible. (Pages 18-32, Lines 338-683)

Minor comments

1. There has been substantial international effort to address issues around depression in the workplace in the past decade. Can the authors comment on whether they think anything has changed in relation to stigma and employment outcomes in the 10 years since this data was collected - would the results still hold if this study was conducted now?

Response: Indeed, in the past 10 years there have been some significant changes, such as changes in the economy, staff shortages in many countries, and more workplace flexibility, which are factors that are beneficial for work participation of people with mental illness. The topic of workplace stigma has received increasing attention over the past two decades but, by far, most studies have been conducted in North America and Europe. The findings of many recent studies suggest that workplace mental health stigma remains a highly understudied and underestimated topic in general given its severe consequences on peoples' overall wellbeing, sustainability of employment and income (read: poverty). The findings of the current study add to these findings illustrating this detrimental effect on people and the need for much more applied research and action, in this respect, especially also in LMICs. We have added a concluding paragraph at the end of the 'Discussion'. (Pages 34-36, Lines 741-772)

2. The free text responses are attached to specific items of the DISC-12, where the items refer to "mental health problem." Were the questions framed to be answered in relation to MDD only, or inclusive of any other mental health condition the participants might have had? This is important to clarify as the authors mention a number of times about this data is novel for MDD in comparison to other disorders including alcohol disorders which are highly comorbid with MDD. Table 1: are there any other employment-related variables from the quantitative survey that could be included to help understand contextual factors of the qualitative data?

Response:

- Re. DISC-12: The DISC (as a mixed-method survey instrument) has a generic reference to "mental health problems" but is meant to be adaptable according to the focus of a specific study (e.g. a study focusing on schizophrenia). In this case, the study focused on people with a diagnoses of Major Depressive Disorder (MDD). The term "mental health problem" in the DISC-12 survey was adapted accordingly for the purpose of this study. Thus, recruited participants with a diagnoses of MDD were asked to answer according to their experiences of stigma and discrimination in relation to their MDD condition. In the open-ended qualitative

responses, participants were invited to give examples of their experiences and to elaborate on their experiences. Our in/exclusion criteria for participants recruitment also explicitly excluded people with co-morbid alcohol and other substance misuse issues. We have added this clarification to the 'Methods' section. (Page 10, Lines 212-213)

- Re. Table 1: Unfortunately, no more information is available.